# The Effect of Oral Iron Supplementation on Gut Microbial Composition: a Secondary Analysis of a Double-Blind, Randomized Controlled Trial among Cambodian Women of Reproductive Age

Emma Finlayson-Trick,[a] Jacob Nearing,[b] Jordie AJ. Fischer,[c,d] Yvonne Ma,[c] Siyun Wang,[c] Hou Krouen,[e] David M. Goldfarb,[a,d,f] Crystal D. Karakochuk[c,d]

[a]Faculty of Medicine, University of British Columbia, Vancouver, British Columbia, Canada

[b]Department of Microbiology and Immunology, Dalhousie University, Halifax, Nova Scotia, Canada

[c]Food, Nutrition and Health, University of British Columbia, Vancouver, British Columbia, Canada

[d]BC Children's Hospital Research Institute, Vancouver, British Columbia, Canada

[e]Helen Keller International, Phnom Penh, Cambodia

[f]Department of Pathology and Laboratory Medicine, BC Children's and Women's Hospital and University of British Columbia, Vancouver, British Columbia, Canada

Emma Finlayson-Trick and Jacob Nearing share co-authorship. Author order was determined alphabetically.

**ABSTRACT** The World Health Organization recommends untargeted iron supplementation for women of reproductive age (WRA) in countries where anemia prevalence is greater than 40%, such as Cambodia. Iron supplements, however, often have poor bioavailability, so the majority remains unabsorbed in the colon. The gut houses many iron-dependent bacterial enteropathogens; thus, providing iron to individuals may be more harmful than helpful. We examined the effects of two oral iron supplements with differing bioavailability on the gut microbiomes in Cambodian WRA. This study is a secondary analysis of a double-blind, randomized controlled trial of oral iron supplementation in Cambodian WRA. For 12 weeks, participants received ferrous sulfate, ferrous bisglycinate, or placebo. Participants provided stool samples at baseline and 12 weeks. A subset of stool samples ($n = 172$), representing the three groups, were randomly selected for gut microbial analysis by 16S rRNA gene sequencing and targeted real-time PCR (qPCR). At baseline, 1% of women had iron-deficiency anemia. The most abundant gut phyla were Bacteroidota (45.7%) and Firmicutes (42.1%). Iron supplementation did not alter gut microbial diversity. Ferrous bisglycinate increased the relative abundance of *Enterobacteriaceae,* and there was a trend towards an increase in the relative abundance of *Escherichia-Shigella*. qPCR detected an increase in the enteropathogenic *Escherichia coli* (EPEC) virulence gene, *bfpA*, in the group that received ferrous sulfate. Thus, iron supplementation did not affect overall gut bacterial diversity in predominantly iron-replete Cambodian WRA, however, evidence does suggest an increase in relative abundance within the broad family *Enterobacteriaceae* associated with ferrous bisglycinate use.

**IMPORTANCE** To the best of our knowledge, this is the first published study to characterize the effects of oral iron supplementation on the gut microbiomes of Cambodian WRA. Our study found that iron supplementation with ferrous bisglycinate increases the relative abundance of *Enterobacteriaceae*, which is a family of bacteria that includes many Gram-negative enteric pathogens like *Salmonella*, *Shigella*, and *Escherichia coli*. Using qPCR for additional analysis, we were able to detect genes associated with enteropathogenic *E. coli*, a type of diarrheagenic *E. coli* known to be present around the world, including water systems in Cambodia. The current WHO guidelines recommend blanket (untargeted) iron supplementation for Cambodian

**Editor** Zhenjiang Zech Xu, 南昌大学

Address correspondence to Crystal D. Karakochuk, crystal.karakochuk@ubc.ca.

The authors declare no conflict of interest.

WRA despite a lack of studies in this population examining iron's effect on the gut microbiome. This study can facilitate future research that may inform evidence-based global practice and policy.

**KEYWORDS** enteropathogen, gut microbiome, iron supplementation

Many gut bacteria are in a constant battle for iron, an essential, growth-limiting nutrient. Iron plays an important role in virulence and colonization for enteropathogens such as *Escherichia coli*, *Salmonella typhimurium*, and *Shigella flexneri* (1, 2). Individuals with excess iron (due to conditions like hemochromatosis) are documented to experience more severe bacterial infections (3–5). Furthermore, *in vitro* and *in vivo* studies have observed a reduction in enteropathogen abundance when iron is limited (6, 7). Given these findings, there is ongoing concern that oral iron supplementation may increase the abundance of iron-dependent enteropathogens.

The WHO recommends untargeted iron supplementation (30 to 60mg iron for 12 weeks) for women in countries where anemia prevalence is greater than 40%. This recommendation is based on the premise that most anemia is due to iron deficiency, which is not always the case. Anemia has many etiologies, including inflammation, infection, hemoglobinopathies, and other micronutrient deficiencies. In Cambodia, for example, the most recent national Demographic and Health Survey found that 45% of women were anemic, but only 3% were iron deficient (8). Genetic hemoglobinopathies (e.g., hemoglobin E type or thalassemia) are prevalent in Cambodia and likely contribute to a greater proportion of the anemia burden than iron deficiency (9). Individuals with severe hemoglobinopathies are already at risk of iron overload due to alterations in iron metabolism; thus, supplementing with iron has the potential to cause further harm (10).

Due to poor bioavailability, iron absorption from oral supplements tends to be low. Simply increasing the dose of iron may not increase the amount absorbed and can lead to gut inflammation causing common gastrointestinal side effects (11). Iron salts, such as ferrous fumarate or sulfate, are frequently used in supplements recommended by the WHO (12, 13). An increasingly favored alternative is ferrous bisglycinate, a stable iron amino acid chelate that, compared to ferrous sulfate, has a two to three times better bioavailability and a more tolerable side effect profile (14). Most clinical trials in this area have focused on characterizing the gastrointestinal side effects associated with iron supplementation (15). Few studies have measured the effects of iron supplements on the gut microbiome and the results have been contradictory (16).

In our study, we used 16S rRNA gene sequencing to examine changes in the gut microbiome following 12 weeks of daily supplementation of either ferrous sulfate or ferrous bisglycinate in non-pregnant Cambodian women of reproductive age (WRA). We then used real-time PCR (qPCR) to assess the presence and abundance of *bfpA* and *eae*, virulence genes from enteropathogenic *E. coli* (EPEC), a common culprit of diarrheal disease worldwide (17–19). Due to the differences in bioavailability among the 2 iron interventions, we hypothesized that the women who received ferrous sulfate would have an increased relative abundance of pathogenic bacteria compared to those who received ferrous bisglycinate.

## RESULTS

**Participant characteristics.** In total, 172 Cambodian women were included in this study (Fig. 1). Baseline characteristics for those included did not significantly differ across trial arms for most reported variables (Table 1). The women had a mean $\pm$ standard deviation (SD) age of 34 $\pm$ 7 years, a mean $\pm$ SD body mass index of 23.5 $\pm$ 3.8, and a median [interquartile range, IQR] number of children of 2 [1, 3]. Among the women, 17% were anemic, defined as hemoglobin (Hb) $<$ 120 g/L. Furthermore, after adjusting for the influence of inflammation, 4% of women were iron deficient (defined by serum ferritin $<$15 $\mu$g/L) and 1% had iron-deficiency anemia (defined as ferritin $<$15 $\mu$g/L and Hb $<$120 g/L) (20). Women were defined as adherent

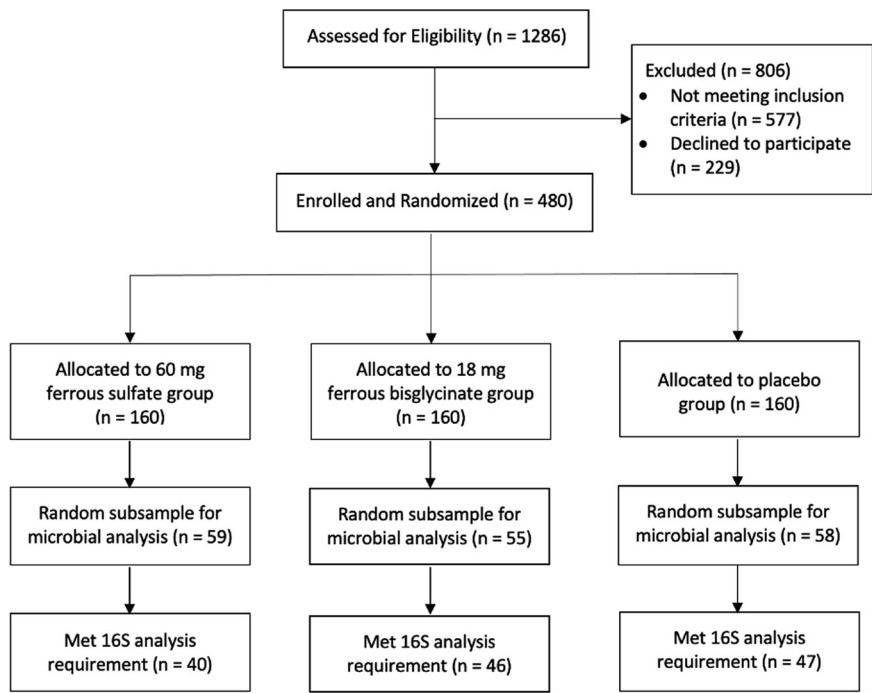

**FIG 1** Flow diagram outlining the allocation of Cambodian women to the different arms of the trial. A subset of stool samples ($n = 172$) were randomly selected for microbial analysis using a computer-generated list ($n = 59$ in the ferrous sulfate group; $n = 55$ for ferrous bisglycinate; $n = 58$ for placebo). Samples were removed following 16S rRNA sequencing if their quality filtered read depth was below 5000 reads (ferrous sulfate $n = 40$; ferrous bisglycinate $n = 46$; placebo $n = 47$).

if they consumed $\geq$ 80% of the capsules, which reflected 62% of women in this study. The one variable that differed between the trial arms was the proportion of women with animals living in the home (Fisher's exact, $P = 0.037$). Overall, phylum composition was comparable to other gut microbiome studies, with communities dominated by Bacteroidota (Fig. S5) (mean 45.7%; SD 23.94%) and Firmicutes (mean 42.1%; SD 20.8%). Other prominent phyla include Proteobacteria (mean 7.5%; SD 11.9%) and Actinobacteriota (mean 4.0%; SD 1.4%).

**Gut microbial diversity in response to iron supplementation.** The change in four different alpha diversity metrics between weeks 0 and 12 were examined for each trial arm, as depicted in Fig. 2. After $P$-value correction, we found no significant differences in alpha diversity change between placebo and either trial arm. Like alpha diversity, we next examined beta diversity and no trail arm showed a change in pairwise distances between week 0 and week 12 that significantly differed from the placebo group (Fig. 3). Further visual examination through principal coordinate analysis plotting and PERMANOVA testing showed no interaction between the week and trial arm for weighted UniFrac (Fig. S6, $P = 0.417$, $r^2 = 0.005$), unweighted UniFrac ($P = 0.905$, $r^2 = 0.004$), or Bray-Curtis dissimilarity ($P = 0.16$, $r^2 = 0.005$). However, a significant effect of sampling week (week 0 versus week 12) was found in both Bray-Curtis dissimilarity ($P = 0.005$, $r^2 = 0.006$) and weighted UniFrac PERMANOVA testing ($P = 0.001$, $r^2 = 0.05$) (Fig. S7).

**Gut microbiome differential abundance analysis.** Using general linear models in ALDEx2, no genera were associated with the interaction between time and trial arm after false discovery correction. Similar results were found using ANCOM-II and MaAsLin2. In comparison, Corncob found three genera (*Enterococcus*, *Weissella*, and *Escherichia-Shigella*) associated with differences in trial arms. Ferrous sulfate was associated with increases in the relative abundance of *Enterococcus* and *Weissella*, while ferrous bisglycinate was associated with increases in *Weissella* and *Escherichia-Shigella* (Fig. 4). Although these significant associations are driven by only a minority of samples. Due to the difficulty of

**TABLE 1** Baseline characteristics of the 172 Cambodian women enrolled in the study divided by treatment arm[a]

| Characteristics | 60 mg ferrous sulfate | 18 mg ferrous bisglycinate | Placebo |
|---|---|---|---|
| Total, n (%) | 59 (34) | 55 (32) | 58 (34) |
| Age, yrs | 32.7 ± 7.1 | 35.3 ± 6.3 | 34.6 ± 8.0 |
| BMI (kg/m$^2$) | 22.9 ± 3.1 | 24.1 ± 4.1 | 23.6 ± 4.2 |
| Parity | 2 [1, 3] | 2 [2, 3] | 2 [1, 3] |
| Household Size | 4.9 ± 1.6 | 4.6 ± 1.6 | 4.4 ± 1.5 |
| | | | |
| Health Centre | | | |
| *Prey Kuy* | 20/57 (35) | 19/57 (33) | 18/57 (32) |
| *Srayov* | 25/64 (39) | 21/64 (33) | 18/64 (28) |
| *Tboung Krapeu* | 14/51 (28) | 15/51 (29) | 22/51 (43) |
| | | | |
| Hemoglobin, g/L | 128.9 ± 11.9 | 127.9 ± 14.0 | 129.2 ± 10.4 |
| Anemia (Hb < 120 g/L) | 7 (12) | 9 (16) | 14 (24) |
| | | | |
| Serum Ferritin,[1] µg/L | 68.3 (45.3, 104.4) | 77.2 (40.0, 105.1) | 62.5 (44.5, 104.1) |
| Iron Deficiency (ferritin < 15 µg/L)[b] | 2 (3) | 2 (4) | 3 (5) |
| Iron Deficiency Anemia (Hb < 120 g/L and ferritin < 15 µg/L)[b] | 1 (2) | 0 (0) | 1 (2) |
| | | | |
| Adherence to supplementation[c] | 30 (51) | 38 (69) | 37 (64) |
| Flush to septic tank household toilet | 56 (95) | 50 (91) | 48 (83) |
| Took antibiotics in last yr | 30 (51) | 18 (33) | 33 (57) |
| | | | |
| Water Source | | | |
| Hand Pump | 27 (46) | 30 (55) | 28 (48) |
| Ringwell | 15 (25) | 12 (22) | 16 (28) |
| | | | |
| Animal(s) living in the home | 51 (86) | 52 (95) | 57 (98) |
| Animal(s) living outside the home | 36 (56) | 35 (64) | 36 (62) |

[a]Values are n (%), mean (SD) or median [IQR]. BMI, body mass index.
[b]Serum ferritin and values were corrected for inflammation. [32].
[c]Women were defined as adherent if they consumed ≥ 80% of the capsules at the week 12 capsule count.

detecting many of the enteropathogens of interest at the genus level, we decided to further test for associations between iron supplementation and the relative abundance of *Enterobacteriaceae* using Corncob. This resulted in a small but significant increase in the relative abundance of this family in the ferrous bisglycinate group compared to the placebo group ($P = 0.026$) (Fig. 5).

**EPEC *bfpA* and *eae*.** qPCR of EPEC genes *bfpA* and *eae* were used to attempt to validate the findings stated above as changes in the relative abundance of *Escherichia-Shigella* showed the strongest relationship to iron supplementation. Across the three trial arms, *bfpA* was detected in 9 to 11% of samples regardless of sample collection week (Table S1). Using mixed effects logistic regression to account for replicate sampling between weeks, a significant association was identified between *bfpA* positivity and the interaction of endpoint sampling and ferrous sulfate trial arm when compared to the placebo group (odds ratio 5.46 [+/− 2.13], $P = 0.0104$) (Table S1). The presence of *eae* was noted in the majority of samples, ranging between 77 and 85% of samples across trial arms, regardless of sample collection week. Mixed effects logistic regression showed no significant interaction between trial arm and sampling time. As a result of the high positivity rate of *eae*, further examination was warranted to assess whether there was an association between the estimated DNA quantity of *eae* and trial arm. The change in estimated log *eae* DNA quantity at week 0 and week 12 for each participant was calculated using a standard curve. No differences in the change of estimated *eae* quantity were found between the different trial arms (Kruskal-Wallis $P = 0.97$) (Fig. S8). Across trial arms, *eae* and *bfpA* were detected in 1.8 to 5.4% of samples at baseline and 2.4 to 4.2% of samples at 12 weeks (Table S2).

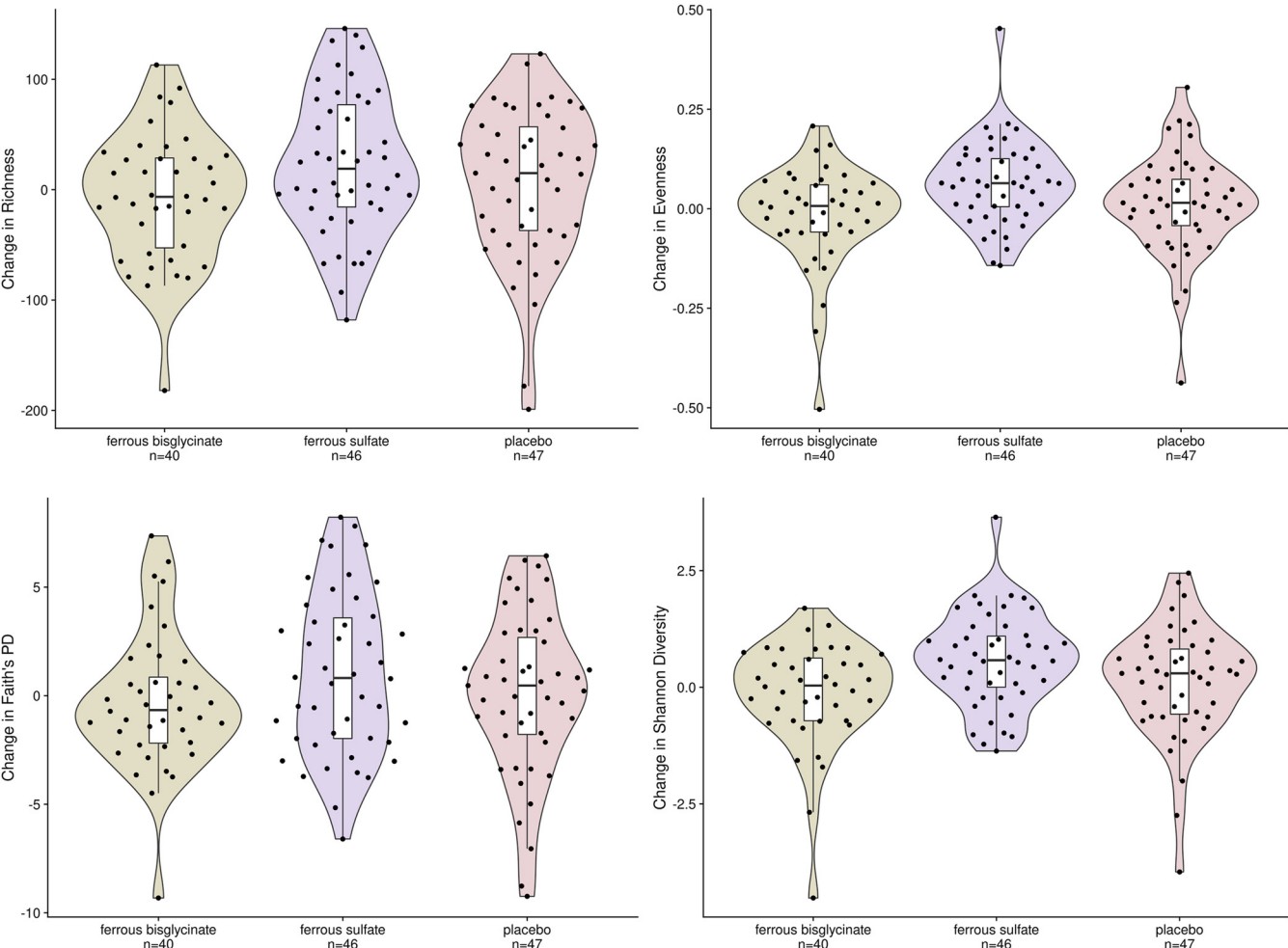

**FIG 2** For each iron supplement trial arm, changes in richness, evenness, Faith's phylogenetic diversity (PD), and Shannon diversity were measured between weeks 0 and 12 and compared to the placebo group using a Wilcoxon Rank Sum Test and a Holm-Bonferroni method with family-wise error rate *P*-value correction. No trail arm showed a change in alpha diversity that was significantly different than the others.

## DISCUSSION

Contrary to our hypothesis, in this study of Cambodian WRA, we found that daily iron supplementation did not significantly affect gut bacterial diversity. We did observe, however, some evidence to suggest that ferrous bisglycinate significantly increased the relative abundance of *Enterobacteriaceae*. We also observed an increase in the relative abundance of *Escherichia-Shigella*, although it should be noted that this association was only detected by one of four differential abundance methods tested on this data set. Nevertheless, Jaeggi and colleagues have previously observed a significant increase in *Escherichia-Shigella* in their population of 6-month-old Kenyan infants following four months of supplementation with iron-enriched micronutrient powder (2.5 mg/12.5 mg iron/day) (21). They also found that infants who received iron supplementation had significantly higher concentrations of pathogenic *E. coli* compared to infants who did not receive iron supplementation. Similarly, Zimmermann et al. (2010) observed a significant increase in *Enterobacteriaceae* among children from Côte d'Ivoire who received iron-fortified biscuits (20 mg iron/day) for six months (22). In contrast, Dostal et al. (2014) did not observe a significant effect on the gut microbiota in 6- to 11-year-old children living in South Africa following 38 weeks of iron supplementation (50 mg iron for 4 days/week) (23). These results highlight that study context is critical as many factors, including hygiene, diet, and parasite burden, can affect gut microbial composition (24, 25). Furthermore, it is important to note that infant and childhood microbiomes are considered to be less stable

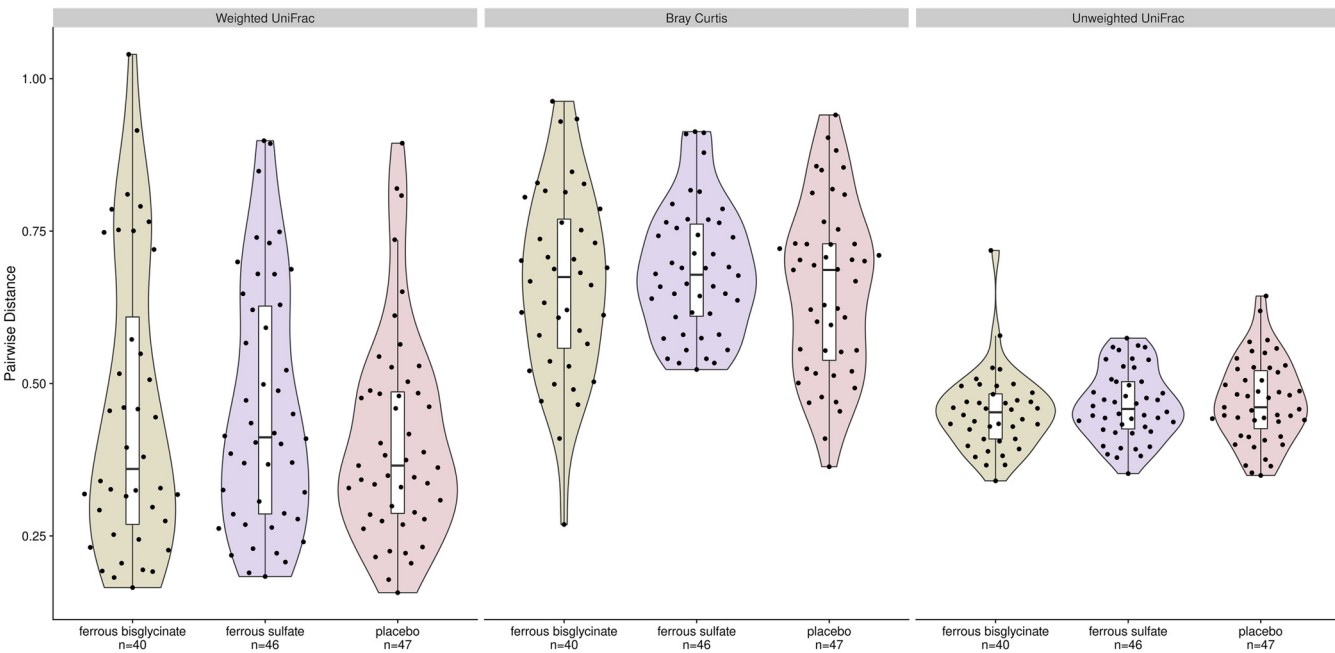

**FIG 3** Weighted UniFrac, Bray-Curtis dissimilarity, and unweighted UniFrac were used for beta diversity analysis to identify any structural changes in the gut microbiome of trial participants. There were no significant changes in the pairwise distances between weeks 0 and 12 for those who received ferrous bisglycinate (*n* = 40), ferrous sulfate (*n* = 46), and the placebo (*n* = 47).

and diverse than adult microbiomes (26). Therefore, our findings need to be confirmed in settings with other non-pregnant WRA.

Diarrheagenic *E. coli* (DEC) are found around the world, including in the waters of Cambodia. In a recent study of Tonle Sap Lake and its tributaries, researchers using qPCR found that 15.3% of *E. coli* isolates were DEC (27). Notably, in nearby floating villages, DEC prevalence changed significantly by season, with a higher prevalence noted in low-water seasons (we will discuss pathogen seasonality further in the paragraph below). DEC includes six *E. coli* pathotypes that are organized based on microbiological and epidemiological characteristics. EPEC, one of the pathotypes, can be further classified based on the *eae* gene, located in the locus of enterocyte effacement, and *bfpA* gene, located on a plasmid called EPEC adherence factor. Typical EPEC strains contain

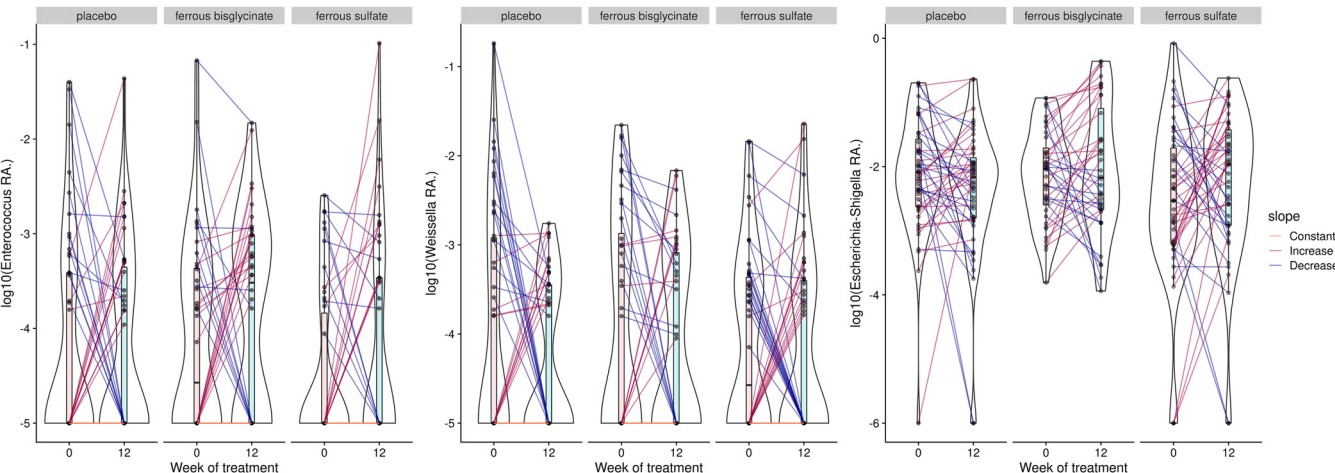

**FIG 4** Microbiome differential abundance tool CornCob identified 3 genera (*Enterococcus*, *Weissella*, and *Escherichia-Shigella*) that responded to iron supplementation. The figure shows average relative abundance. Ferrous sulfate increased the relative abundance of *Enterococcus* and *Weissella*, while ferrous bisglycinate increased *Weissella* and *Escherichia-Shigella*. Colors were used to indicate direction of change with orange indicating no change ("constant"), red indicating increase, and blue indicating decrease.

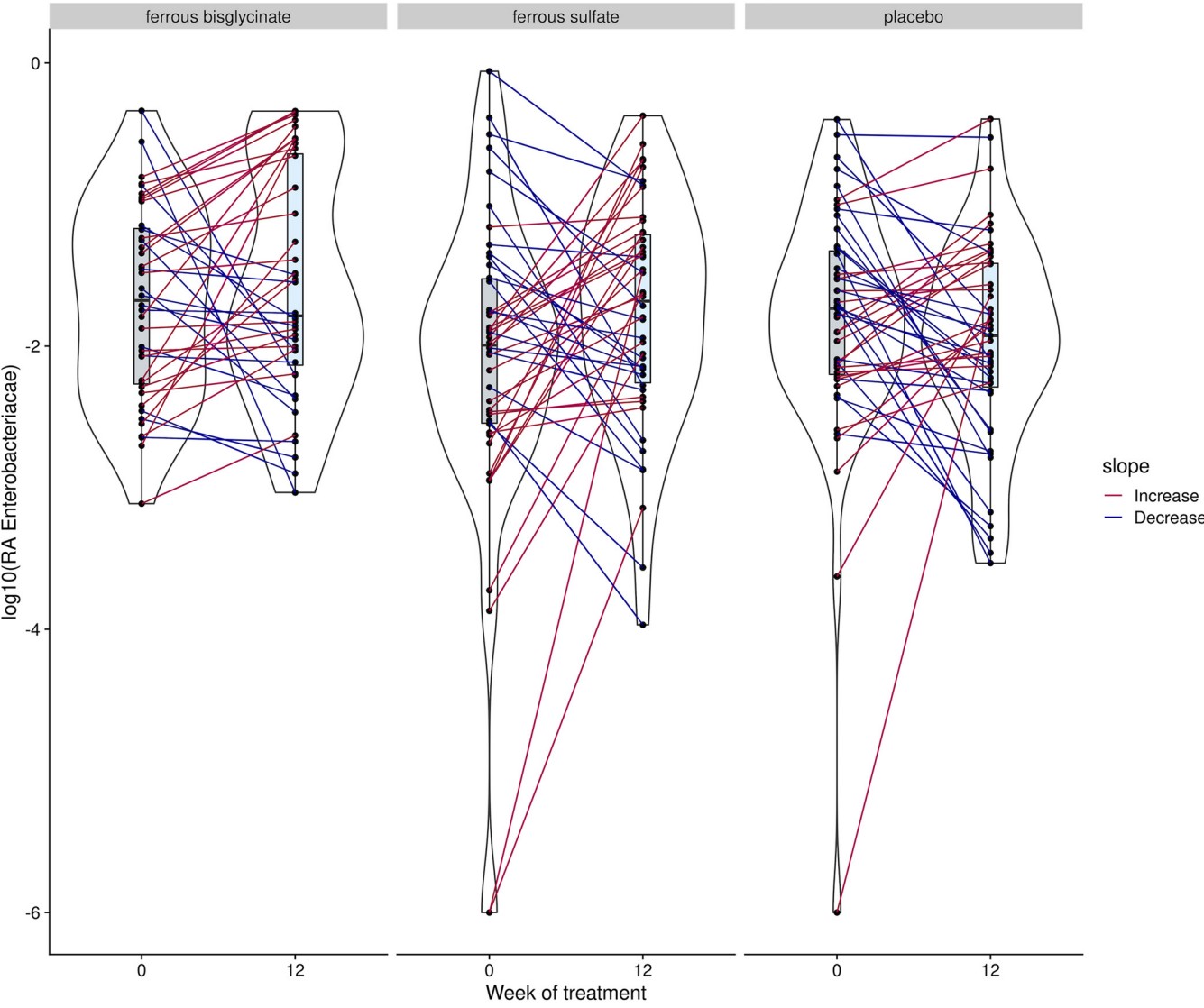

**FIG 5** The relative abundance of *Enterobacteriaceae* showed a small but significant increase following the 12-week trial with ferrous bisglycinate. Colors were used to indicate direction of change with red indicating increase and blue indicating decrease.

*eae* and *bfpA*, whereas atypical EPEC strains contain *eae*, but are missing *bpfA*. Our qPCR results, which used primers specific for *E. coli*, suggest that the majority of our samples positive for *eae* were atypical EPEC; however, it should be noted that entero-hemorrhagic *E. coli* can also contain *eae* (27). As we did not examine the other genes associated with enterohemorrhagic *E. coli*, we are unable to comment on the preva-lence of this DEC pathotype within our population. Nevertheless, several studies world-wide have documented the role of both typical and atypical EPEC in diarrheal disease, although typical EPEC generally has a stronger association with development of diar-rheal disease (28–31). In the main trial population for this study, 53% of women ($n = 254/480$) reported experiencing gastrointestinal upset at least once a month at baseline, with 26% ($n = 67/254$) of these women experiencing diarrhea (defined as three or more loose bowel movements in 24 h, citing data from a primary manuscript under review at the *Journal of Nutrition*). At 12 weeks, 17% ($n = 73/441$) of women reported adverse side effects, of these, 7% ($n = 5/73$) reported diarrhea.

Cambodia is an equatorial country with a dry and rainy season that can be further categorized into pre- and post-monsoon seasons (32). Due to annual droughts and flooding, waterborne diarrheal diseases are a concern in Cambodia, especially in rural

regions (33). Drought is thought to concentrate pathogens within remaining water sources and has globally been linked to increases in salmonellosis, shigellosis, and leptospirosis (34). Similar to the study out of Tonle Sap Lake, Poirot and colleagues observed higher concentrations of *E. coli* during the dry season in water sources collected from three Cambodian provinces (35). Our study occurred primarily during the dry season (January/February), and all study sites were located in rural Kampong Thom province.

In addition to seasonality, this study must also consider the impact of COVID-19, which was declared a global pandemic by the WHO on March 11th, 2020. Cambodia rapidly implemented stringent measures to slow the spread of COVID-19 (36). While research continues to elucidate the direct and indirect influence of COVID-19 on the gut microbiome, it is understood that gut microbial richness is a balance between microbial acquisition and loss (37). Many public health measures have indirectly prevented acquisition (e.g., limited contact between people) and promoted loss (e.g., hand hygiene) of gut microbial species (38). The implementation of public health guidelines, which occurred after baseline collection and before 12-week collection, may aid in explaining why the placebo group also experienced a similar change in gut microbial diversity over the 12 week trial (Fig. 3).

We have identified three limitations of our study. Firstly, stool does not necessarily accurately represent the composition and metagenomic function of mucosa-associated microbiota (39). In order to document this population of microbes, we would need to perform colonoscopies, which are not feasible for large studies. As such, until the advent of new sampling methods, such as swallowable devices, it will remain unclear how mucosa-associated microbiota respond to iron supplementation (40, 41). Secondly, our study used randomly sampled stool. As described by Hsieh and colleagues, stool preparation (randomly sampled versus homogenized) does not impact intra-individual bacterial diversity. Still, there are differences at the level of individual taxa, including higher proportions of *Faecalibacterium* and decreased proportions of *Bacteroides* (42). As participants randomly collected their stool, our results may represent an over or under-sampling of specific bacterial taxa (43). As many bacteria were not detected in our samples during 16S rRNA gene sequencing, future studies may test for specific pathogens such as *Salmonella*, *Escherichia*, *Campylobacter*, *Vibrio*, and *Plesiomonas*. Finally, we acknowledge that there were only two collection time points during our 12 week trial. These two time points allowed us to record overall change versus any rapid fluctuations, which may be of interest in future research (21).

To our understanding, this is the first study to examine the effect of iron supplementation on the gut microbiome in non-pregnant WRA. We observed that iron supplementation, in the form of ferrous bisglycinate, increases the relative abundance of *Enterobacteriaceae*. Currently, the WHO recommendations suggest that women in Cambodia receive daily iron supplementation for three months; however, besides a small number of studies, there is little evidence examining the possible harms of this regimen. We hope the present study provides a foundation for future research to ensure policymakers and Cambodian WRA have the necessary information to guide iron supplementation practices.

## MATERIALS AND METHODS

**Study design and eligibility criteria.** This study was nested within a larger randomized controlled trial of oral iron supplementation that included 480 non-pregnant Cambodian WRA from the rural Kampong Thom province. The primary trial aimed to assess the non-inferiority of 18 mg iron as ferrous bisglycinate compared with 60 mg iron as ferrous sulfate on mean ferritin concentrations and other biomarkers following 12 weeks daily supplementation.

Study inclusion criteria involved healthy, non-pregnant women 18 to 45 years old who consented to participate in the study by providing blood and stool samples and resided in the study location for the entire study period. The women lived in villages within the following health center catchment areas: Prey Kuy, Tboung Krapeu, and Srayov. Participants were excluded if they had any known illness or disease, were pregnant, or were taking antibiotics, non-steroidal anti-inflammatory drugs, dietary supplements, or vitamin and mineral supplements in the previous 12 weeks. Women were recruited by convenience sampling (rolling recruitment began in December 2019), and if eligible, they were enrolled once they provided written consent. Women were randomized at the health center level with a 1:1:1 allocation ratio to either

ferrous sulfate, ferrous bisglycinate, or placebo ($n = 160$ in each trial group). The ferrous sulfate capsule contained 60 mg of elemental iron (standard treatment), the ferrous bisglycinate capsule contained 18 mg of elemental iron (experimental treatment), and the placebo capsules contained microcrystalline cellulose (no elemental iron). The capsules were identical in composition, taste, smell, and appearance. Women provided a neat stool sample using an at-home stool collection kit at baseline (five-week rolling enrollment: January to February 2020) and after 12 weeks of daily supplementation (April to May 2020) (Fig. S1) (43). Information, such as gastrointestinal indicators, were captured in baseline and 12-week questionnaires. Regular monitoring visits were made by local research staff to participants' homes to ensure daily supplementation adherence and record reported side effects. Trial investigators, research staff, and participants were blinded to the assigned trials. The full trial protocol has been reported elsewhere (44).

**Stool collection.** Stool samples were placed on ice and transported for 4 to 6 h to the National Institute of Public Health Laboratory in Phnom Penh, Cambodia and were immediately frozen at $-20°C$. They were later shipped on dry ice to the University of British Columbia, Vancouver, Canada laboratory and were frozen upon receipt at $-80°C$ until analysis.

**Laboratory analyses.** A subset of stool samples ($n = 172$) was randomly selected for microbial analysis using a computer-generated list ($n = 59$ received ferrous sulfate; $n = 55$ received ferrous bisglycinate; $n = 58$ received placebo).

**(i) DNA extraction and 16S rRNA gene sequencing.** DNA was extracted using a Mobio PowerFecal extraction kit (Qiagen) according to the manufacturer's protocol. DNA purity was assessed using a Nanodrop spectrophotometer (NanoDrop Technologies Inc.). DNA samples were submitted for 16S rRNA gene sequencing at the Integrated Microbiome Resource (IMR) at Dalhousie University. The IMR provided 16S rRNA gene primers (B969F and BA1406R). Variable regions V6 to V8 of the bacterial 16S rRNA gene were amplified from all purified DNA samples using 25 cycles of PCR and Phusion+ high-fidelity polymerase. Amplified DNA was then sequenced on an Illumina MiSeq using paired-end 300 bpsequencing.

**(ii) Quantitative real-time PCR (qPCR).** Target genes associated with EPEC, *bfpA*, and *eae*, were amplified from extracted DNA using SSoAdvanced Universal SYBR green Supermix (Bio-Rad Laboratories Inc.) on a CFX96 Touch Real-Time PCR Detection System (Bio-Rad Laboratories Inc). The primers were designed by Cabal and colleagues and are listed here: bfpA-F (CMGGTGTGATGTTTTACTAC), bfpA-R (TGCCCAATATACARACCAT), eae-F (GCTATAACRTCTTCATTGATC), and eae-R (RCTACTTTTRAAATAGTCTCG) (Integrated DNA Technologies) (18, 19). DNA extraction was performed on EPEC strain E2348/69 using the PrepMan Ultra Sample Preparation Reagent (Thermo Fisher Scientific) for use as a positive control for the *bfpA* and *eae* genes (18, 19). The manufacturer's protocol was modified as follows: initial denaturation at 95°C for 3 min, followed by 40 cycles of 95°C for 15 s, and 52°C for 30 s, followed by a melt-curve analysis from 65°C to 95°C. All samples were run in triplicate, and the data were analyzed using the CFX Maestro V.4.1.2433.1219 (Bio-Rad Laboratories Inc).

**Data analyses. (i) 16S sequence processing.** Paired-end reads of V6-V8 amplification primers were trimmed off using cutadapt with default settings (45). Forward and reverse primer trimmed reads were stitched together using the QIIME2 (v. QIMME2-2020.8) VSEARCH join-pairs plugin (46, 47). Stitched reads were quality filtered using the QIIME2 q-score plugin with default settings. The QIIME2 plugin Deblur was used to group filtered primer-free stitched reads into amplicon sequence variants using a trim length of 399 bp and a minimum read requirement of one (48). The mean sequencing depth across samples was then calculated using QIIME2, and amplicon sequence variants (ASVs). ASVs with read counts lower than 0.1% across all samples were removed. ASVs were then placed into the Greengenes 13_8 99% reference 16S rRNA *gene* tree using the QIIME2 plugin fragment-insertion SEPP (49–51).

**(ii) Microbiome 16S diversity analysis.** Rarefaction curves were generated using the QIIME2 diversity alpha-rarefaction visualizer (Fig. S2). A sequencing depth of 5000 reads was chosen based on the depth at which microbial richness plateaued. Diversity metrics were generated using QIIME2 with a rarefaction depth of 5000 reads and the previously generated phylogenetic tree. Changes in richness, evenness, Shannon diversity, and Faith's phylogenetic diversity were then measured between weeks 0 and 12 and compared to the placebo group for each iron supplement group using a Wilcoxon Rank Sum Test. *P*-values were then corrected using the Holm-Bonferroni method to help control for family-wise error rates.

Three different beta diversity metrics were compared to identify any structural changes in the gut microbiome of trial participants. These metrics were weighted UniFrac, unweighted UniFrac, and Bray-Curtis dissimilarity. Changes in pairwise distances/dissimilarities between week 0 before supplementation and week 12 after supplementation were compared using Wilcoxon Rank Sum Test. For each distance/dissimilarity matrix, we conducted PERMANOVA testing, examining the interaction between trial arm and week of sample collection, using the adonis2 function in the vegan R package. We additionally accounted for our repeated sampling design by setting block permutations to subject ID. Furthermore, each matrix was also used in principal coordinates analysis to visualize any potential clustering by trial arm and week of sample collection.

**(iii) Microbiome 16S differential abundance.** ASVs were assigned taxonomy using a QIIME2 naive Bayes classifier trained on the 99% Silva V138 database (52). Samples below 5000 reads were filtered out, and read counts were converted to either relative abundances or centered log-ratios with a pseudo count of 1. After filtering out samples with less than 5000 reads and those without a pair (missing either baseline or endline), 40 sample pairs remained in the ferrous sulfate group, 46 in the ferrous bisglycinate group, and 47 in the placebo group. These samples were then examined for differential microbial abundance using four separate tools: ALDEx2, Corncob, MaAsLin2, and ANCOM-II (53–56). These bioinformatic tools were chosen based on previous performance analysis and the ability to test methods that were compositionally aware (ALDEx2 and

ANCOM-II) or tested for differences in microbial proportions (Corncob, MaAsLin2) (57). Time and trial groups were assessed for significant associations with genus level abundances. During modeling, the time reference group was set to the baseline collection and the trial reference group was placebo. After differential abundance testing, any genera or ASVs found in less than 5% of samples across the data set were removed from consideration before multiple test corrections.

ALDEx2 was run using 128 Monte Carlo samplings and generalized linear modeling using default Benjamin-Hochberg false discovery correction. The interaction between time and trial arm was examined for significance. Corncob was run by combining data into a phyloseq object and then running the differentialTest function using a Wald test and controlling for differential variability in the interaction between trial arm and week of collection (58). MaAsLin2 was run using default settings and an arcsine square-root transformation. Interactions were modeled using artificial interaction columns generated in R as described in the MaAsLin2 tutorial. Finally, ANCOM-II was run by first identifying structural zeros using the function feature_table_pre_process. Structural zeros and metadata were then passed into the main ANCOM function using the interaction between trial arm and time as the "main_var" and group + time as the "adj_formula".

**(iv) qPCR Analysis.** Of the 172 baseline-endline pairs, 166 baseline, and 166 endline samples were used for qPCR analysis (some samples were removed due to insufficient material). The standard curve formula for *bfpA* ($r^2 = 0.9959$) was applied to the Cq values obtained from each sample (Fig. S3). The *bpfA* gene was not present in every sample; therefore, presence-absence analysis was conducted using mixed effects logistic regression, examining the interaction between time and trial arm with a random subject effect. The *eae* gene was present in most samples; therefore, estimated DNA quantities were examined by first constructing a standard curve from positive controls. The standard curve formula for *eae* ($r^2 = 0.9856$) was then applied to the Cq values obtained from each sample (Fig. S4). For the purposes of estimated DNA quantity analysis, we set the log DNA quantity of samples lacking a Cq value as the detection limit measured on positive control samples. Changes in log *eae* quantity (ng) were then calculated for each sample between weeks 0 and 12. This change was then examined across trial arms using an ANOVA.

**Ethics approval and consent to participate.** Ethical approval was obtained from the University of British Columbia Clinical Research Ethics Board, Canada (H15-00933, 19 June 2015) and the National Ethics Committee for Health Research, Cambodia (110-NECHR, 24 April 2015) and was registered at clinicaltrials.gov (NCT04017598).

**Data and materials availability.** The data sets generated and/or analyzed during the current study are available in the ENA repository, and can be found at the following accession number ERP138269 (https://www.ebi.ac.uk/ena/browser/view/PRJEB53465?show=reads). Metadata tables, qPCR data, and code used to produce the results within this study can be found at https://github.com/nearinj/Cambodia_Iron_Supplement.

## SUPPLEMENTAL MATERIAL

Supplemental material is available online only.

**SUPPLEMENTAL FILE 1**, PDF file, 0.6 MB.

## ACKNOWLEDGMENTS

We thank Ngik Rem, and Helen Keller International (Cambodia) for managing and supervising study operations; and Chanthan Am, National Institute of Public Health Laboratory (Cambodia) for overseeing sample processing in Phnom Penh. We thank The Factors Group of Nutritional Companies (Burnaby, BC, Canada) for manufacturing the study supplements. We thank André Comeau from the Integrated Microbiome Resource at Dalhousie University for guiding the 16S rRNA gene sequencing.

We declare that we have no competing interests.

E.F.-T. is supported by the Association of Medical Microbiology and Infectious Disease in the form of the Canada Medical Student Research Award. J.N. is supported by researchNS and the Nova Scotia Graduate Scholarship. C.D.K. is supported by a Michael Smith Foundation for Health Research Scholar Award and a Canada Research Chair Tier 2 in Micronutrients and Human Health. J.A.J.F. received student funding from the Canadian Institutes of Health Research. E.F.-T., J.A.J.F., D.M.G. and C.D.K. designed the research; J.A.J.F and H.K. oversaw trial implementation and data collection in Cambodia; E.F.-T. performed DNA extraction and sequencing preparation; J.N. performed the 16S RNA sequencing analysis; Y.M. and S.W. performed the qPCR analyses; E.F.-T., J.A.J.F., and J.N. conducted the statistical analyses and drafted the manuscript; all authors contributed to interpretation of the results; C.D.K. had primary responsibility for content; and all authors contributed to writing and manuscript revision and read and approved the final manuscript.

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
