## [Reviewer comments · Microbiology Spectrum]

Microbiology Spectrum

The effect of oral iron supplementation on gut microbial composition: A secondary analysis of a double-blind, randomized controlled trial among Cambodian women of reproductive age

Emma Finlayson-Trick, Jacob Nearing, Jordie Fischer, Yvonne Ma, Siyun Wang, Hou Krouen, David Goldfarb, and Crystal Karakochuk

Corresponding Author(s): Crystal Karakochuk, University of British Columbia

Review Timeline:

Submission Date:	December 22, 2022
Editorial Decision:	February 25, 2023
Revision Received:	April 19, 2023
Accepted:	April 28, 2023

Editor: Zhenjiang Xu

Reviewer(s): Disclosure of reviewer identity is with reference to reviewer comments included in decision letter(s). The following individuals involved in review of your submission have agreed to reveal their identity: Loo Wee Chia (Reviewer #2); Scott G. Daniel (Reviewer #3)

Transaction Report:

DOI: <https://doi.org/10.1128/spectrum.05273-22>

February 25, 2023

Dr. Crystal D Karakochuk
University of British Columbia
Food, Nutrition, and Health
Vancouver
Canada

Re: Spectrum05273-22 (The effect of oral iron supplementation on gut microbial composition: A secondary analysis of a double-blind, randomized controlled trial among Cambodian women of reproductive age)

Dear Dr. Crystal D Karakochuk:

Link Not Available

Sincerely,

Zhenjiang Xu

Journals Department
Reviewer comments:

Reviewer #1 (Comments for the Author):

The manuscript entitled "The effect of oral iron supplementation on gut microbial composition: A secondary analysis of a double-blind, randomized controlled trial among Cambodian women of reproductive age" by Finlayson-Trick & Nearing et al studied the effect of iron supplementation on gut microbiome in 172 Cambodian women. The study design and the choice of the participants seem pertinent as well as the choice of the multiple statistical analyses performed by the authors. Overall, the authors could show that iron supplementation did not significantly alter gut microbial diversity. Nevertheless, small changes in the abundance of some bacterial genera were measured in response to iron supplementation. The authors propose that iron supplementation could cause the increase in enteropathogenic bacteria and harm the population treated. This study is of importance as WHO

recommends iron supplementation without knowing its repercussion(s) on the risk of infection.

Major:

1/ The authors measured a significant increase in Enterobacteriaceae upon ferrous bisglycinate supplementation while detecting an increase in bfpA positivity mainly in the case of ferrous sulfate supplementation. Could the author discuss this discrepancy? Also, could they be more specific about which of these two results seems to be the most compelling? Also, while the authors study two types of iron supplementations, could they conclude or at least comment on which supplementation seem the most effective?

2/ Would it be possible that the quantification of genes by PCR is dependent on the overall number of bacteria in the stool sample? How was normalized the detection of genes such as bfpA?

Minor:

-In Figure 4, the colors displayed do not correspond to the legend of the figure (Purple shows increase, not decrease and orange shows decrease not increase). A better quality of the image as well as a better color choice would help the reader decrypt the figure presented.

-L328: Change "whereas atypical EPEC strains are contain eae" to "whereas atypical EPEC strains contain eae"

Reviewer #2 (Comments for the Author):

The manuscript "The effect of oral iron supplementation on gut microbial composition: A secondary analysis of a double-blind, randomized controlled trial among Cambodian women of reproductive age" by Finlayson-Trick et al. investigated the impact of oral iron supplementations on gut microbiota. This is an important area of research, laying the groundwork for evidence-based guidance around the management of iron deficiency anemia. However, the manuscript provides limited insight and lacks of several key microbiota analysis such as compositional analysis at different taxonomic levels.

General comments:

- 16S rRNA V6-V8 regions were targeted for sequencing instead of the commonly used V4-V5 regions. Any justification for the choice taken?
- Around 10 subjects were removed per study arm due to poor sequencing outcome. What could be the contributing factor?
- Figure 4 & 5, please include statistical analysis.
- Please include compositional analysis of gut microbiota at different taxonomic levels.
- It would be interesting to investigate the correlation between gut microbiota and the occurrence of gastrointestinal symptoms.
- (Line 301 - 303): 'We did observe, however, some evidence to suggest that ferrous bisglycinate significantly increased the relative abundance of Enterobacteriaceae, a finding consistent with other studies in this field'. Are there any references related to ferrous bisglycinate?
- The dosage of elemental iron differs greatly between the supplemental forms. How much of iron could end up in the lower gastrointestinal tract?

Specific comments:

- Figure 3: Please simplify the legend for figure B, merging the representation for week and sample type.

Reviewer #3 (Comments for the Author):

This article is well written and supports the conclusions of the study. I appreciate that the authors published what is essentially negative results and made available all data and code. However, there are several changes that should be made to improve the clarity of the conclusions. Specifically:

Lines 37-42: Remove all instances of "significantly". If something significantly increases, just say increases. If something doesn't significantly increase, then it is OK to say trending. "Significantly" is over-used in scientific literature and unnecessary.

Line 44: Remove "appears to".

Line 57: Instead of "we hope" use language like This study can facilitate...

Line 98-99: Just say, "and the results have been contradictory". No need for "of those that have".

For Fig. 2: This is a confusing figure. It would make more sense to me if the weeks would be shown alongside each other rather than having "Change in X" on the y-axis (and it should be "Change in X between weeks 0 and 12"). Stars should be explained in

the legend. I'm not sure why you compared the time changes between trial arms if you didn't mention it in the text (or the Kruskal-Wallis test for that matter).

Fig. 3: "Sample Type" in legend should be something like "Trial arm and Week"

Fig. 4: This is a busy figure and I'm having a hard time understanding why you chose to plot like this. The legend says "Ferrous sulfate increased the relative abundance of Enterococcus and Weissella" but for Weissella it looks like 15 increased while 20 decreased. I'm assuming you mean the average abundance increased? Also, it looks like the colors are switched. The orange lines look like they are decreasing and vice versa for purple. And for Shigella, the green lines definitely have slopes so I don't understand why they would be considered constant. I also don't understand the inclusion of the tables unless you were doing some kind of chi-square or Fisher test.

Fig. 5: Make the color scheme consistent with Fig. 4.

Staff Comments:

Preparing Revision Guidelines

Please return the manuscript within 60 days; if you cannot complete the modification within this time period, please contact me. If you do not wish to modify the manuscript and prefer to submit it to another journal, please notify me of your decision immediately so that the manuscript may be formally withdrawn from consideration by Microbiology Spectrum.

Dear Editor,

The manuscript "The effect of oral iron supplementation on gut microbial composition: A secondary analysis of a double-blind, randomized controlled trial among Cambodian women of reproductive age" by Finlayson-Trick et al. investigated the impact of oral iron supplementations on gut microbiota. This is an important area of research, laying the groundwork for evidence-based guidance around the management of iron deficiency anemia. However, the manuscript provides limited insight and lacks of several key microbiota analysis such as compositional analysis at different taxonomic levels.

General comments:

- 16S rRNA V6-V8 regions were targeted for sequencing instead of the commonly used V4-V5 regions. Any justification for the choice taken?
- Around 10 subjects were removed per study arm due to poor sequencing outcome. What could be the contributing factor?
- Figure 4 & 5, please include statistical analysis.
- Please include compositional analysis of gut microbiota at different taxonomic levels.
- It would be interesting to investigate the correlation between gut microbiota and the occurrence of gastrointestinal symptoms.
- (Line 301 - 303): 'We did observe, however, some evidence to suggest that ferrous bisglycinate significantly increased the relative abundance of Enterobacteriaceae, a finding consistent with other studies in this field'. Are there any references related to ferrous bisglycinate?
- The dosage of elemental iron differs greatly between the supplemental forms. How much of iron could end up in the lower gastrointestinal tract?

Specific comments:

- Figure 3: Please simplify the legend for figure B, merging the representation for week and sample type.

Lines 37-42: Remove all instances of "significantly". If something significantly increases, just say increases. If something doesn't significantly increase, then it is OK to say trending.

"Significantly" is over-used in scientific literature and unnecessary.

Line 44: Remove "appears to".

Line 57: Instead of "we hope" use language like This study can facilitate...

Line 98-99: Just say, "and the results have been contradictory". No need for "of those that have".

For Fig. 2: This is a confusing figure. It would make more sense to me if the weeks would be shown alongside each other rather than having "Change in X" on the y-axis (and it should be "Change in X between weeks 0 and 12"). Stars should be explained in the legend. I'm not sure why you compared the time changes between trial arms if you didn't mention it in the text (or the Kruskal-Wallis test for that matter).

Fig. 3: "Sample Type" in legend should be something like "Trial arm and Week"

Fig. 4: This is a busy figure and I'm having a hard time understanding why you chose to plot like this. The legend says "Ferrous sulfate increased the relative abundance of Enterococcus and Weissella" but for Weissella it looks like 15 increased while 20 decreased. I'm assuming you mean the average abundance increased? Also, it looks like the colors are switched. The orange lines look like they are decreasing and vice versa for purple. And for Shigella, the green lines definitely have slopes so I don't understand why they would be considered constant. I also don't understand the inclusion of the tables unless you were doing some kind of chi-square or Fisher test.

Fig. 5: Make the color scheme consistent with Fig. 4.

Spectrum Reviewer Comments

We would first like to thank the reviewers and editors for their time and contributions toward our manuscript. The critical feedback we have received has been considered and substantially improved our manuscript. Below we have noted our comments for each point the reviewers have made and the actions we have taken to address them.

Reviewer #1

The manuscript entitled "The effect of oral iron supplementation on gut microbial composition: A secondary analysis of a double-blind, randomized controlled trial among Cambodian women of reproductive age" by Finlayson-Trick & Nearing et al. studied the effect of iron supplementation on gut microbiome in 172 Cambodian women. The study design and the choice of the participants seem pertinent as well as the choice of the multiple statistical analyses performed by the authors. Overall, the authors could show that iron supplementation did not significantly alter gut microbial diversity. Nevertheless, small changes in the abundance of some bacterial genera were measured in response to iron supplementation. The authors propose that iron supplementation could cause the increase in enteropathogenic bacteria and harm the population treated. This study is of importance as WHO recommends iron supplementation without knowing its repercussion(s) on the risk of infection.

Major:

1. The authors measured a significant increase in *Enterobacteriaceae* upon ferrous bisglycinate supplementation while detecting an increase in *bfpA* positivity mainly in the case of ferrous sulfate supplementation. Could the author discuss this discrepancy? Also, could they be more specific about which of these two results seems to be the most compelling? Also, while the authors study two types of iron supplementations, could they conclude or at least comment on which supplementation seem the most effective?

These discrepant findings are interesting and we have several comments to address this: First, by visually inspecting the relationships in our differential abundance analysis, we note that the effect size was relatively small and that our result with *Enterobacteriaceae* was driven by a minority of samples. Ultimately, we were limited by a small sample size in this study. Secondly, it is critical to note that these discrepant findings represent two very different scopes of analysis, namely a bacterial family and a specific bacterial gene. Additionally, the techniques used to generate these results (e.g., 16S rRNA sequencing and qPCR) have their own inherent biases, which makes comparison of these two results very challenging. Nevertheless, it is possible that within the family of *Enterobacteriaceae* there are a few species that do not have *bfpA*, but who are driving the response to ferrous bisglycinate. Given the limitations stated above and in our manuscript (lines 392-407), it is challenging to state which result is the most compelling. Other studies in this field have similarly found increases in *Enterobacteriaceae* following iron supplementation (Zimmerman et al. 2010).

In terms of effectiveness, mean ferritin concentrations (95% CI) at 12 weeks was higher in the ferrous sulfate (99 [95, 103] µg/L, p<0.001) than the ferrous bisglycinate (84 [80, 88] µg/L) and placebo groups (78 [74, 82] µg/L). Further, only 17% of women (73/441) reported any adverse side effects at 12 weeks (at the end of the trial); the proportion of women who reported any adverse side effects at 12 weeks did not differ by intervention group (chi-square, p=0.72). These

are the main trial findings reported in our primary manuscript which is currently under review at *The Journal of Nutrition* [revisions were submitted in early April]. The citation for the main trial manuscript will be updated in the text accordingly, upon acceptance for publication.

2. Would it be possible that the quantification of genes by PCR is dependent on the overall number of bacteria in the stool sample? How was normalized the detection of genes such as *bfpA*?

The quantification of genes is dependent on the number of bacteria in the stool sample, so if there is more EPEC in the sample, there are more copies of the DNA detected. We extracted DNA from the same amount of stool between all the samples, which is standard practice and allows for fair comparison. We then normalized our samples against the positive DNA control and all the runs used the same volume and concentration of DNA for the positive sample.

Minor:

3. In Figure 4, the colors displayed do not correspond to the legend of the figure (Purple shows increase, not decrease and orange shows decrease not increase). A better quality of the image as well as a better color choice would help the reader decrypt the figure presented.

We concur and have now included a high quality image with a revised legend (the colours now correspond with the legend).

4. L328: Change "whereas atypical EPEC strains are contain eae" to "whereas atypical EPEC strains contain eae"

We removed the word "are" from L328.

Reviewer #2

The manuscript "The effect of oral iron supplementation on gut microbial composition: A secondary analysis of a double-blind, randomized controlled trial among Cambodian women of reproductive age" by Finlayson-Trick et al. investigated the impact of oral iron supplementations on gut microbiota. This is an important area of research, laying the groundwork for evidence-based guidance around the management of iron deficiency anemia. However, the manuscript provides limited insight and lacks several key microbiota analysis such as compositional analysis at different taxonomic levels.

General comments:

1. 16S rRNA V6-V8 regions were targeted for sequencing instead of the commonly used V4-V5 regions. Any justification for the choice taken?

We concur that V4-5 is commonly considered the universal target as it has excellent coverage for bacterial, archaeal, and eukaryotic samples. Nevertheless, in their manuscript entitled "Microbiome Helper: a Custom and Streamlined Workflow for Microbiome Research", Comeau et al. (2017) observed that the V4-5 region overrepresents *Firmicutes* and *Bacteroides* while severely underestimating *Actinobacteria* and *Propionibacterium*. In comparison, the V6-8 region showed more accurate proportions of *Actinobacteria* and *Firmicutes* but overestimated *Proteobacteria*. As such, we decided to sequence the V6-8 region in consultation with the Dalhousie University Integrated Microbiome Resource (<https://imr.bio/protocols.html>) as it is generally recommended not to use V4-5 to measure bacterial diversity if the sample contains substantial eukaryote "host/associated" contamination.

2. Around 10 subjects were removed per study arm due to poor sequencing outcome. What could be the contributing factor?

Due to natural variations in read count across samples, rarefaction is used to determine whether a specific sample has been sufficiently sequenced to represent its identity. This way all samples will have the same number of reads, which allows for better comparison of diversity metrics. Based on our rarefaction curve (Fig. S2), we selected a sequencing depth of 5000 reads. Consequently, we removed approximately 10 subjects per study arm who did not meet this read cut off. A recent publication by Celis et al. (2022) entitled “Optimization of the 16S rRNA sequencing analysis pipeline for studying *in vivo* communities of gut commensals” provides further insight into this process. It is important to note that the same DNA extraction kit was used for all our samples, and we quantified our extracted DNA to ensure that approximately the same concentration was sent for sequencing.

3. Figure 4 & 5, please include statistical analysis.

The data presented in Figure 4 and 5 are the statistically significant results generated through our differential abundance analysis. We have removed the tables to avoid confusion.

4. Please include compositional analysis of gut microbiota at different taxonomic levels.

In our current text we provide differential abundance data at the genera and family level (Figure 4 and 5, respectively). We have added a stacked bar chart showing the relative abundance of each phyla within the samples organized by trial arm and week (Supplemental Figure 5).

5. It would be interesting to investigate the correlation between gut microbiota and the occurrence of gastrointestinal symptoms.

We agree that this is an area of interest. In the manuscript reporting the main trial findings that is currently under review at *The Journal of Nutrition*, we report the adverse side effects reported among the participants. In summary, only 17% of women (73/441) reported any adverse side effects at 12 weeks (at the end of the trial); the proportion of women who reported any adverse side effects at 12 weeks did not differ by intervention group (chi-square, $p=0.72$). It was not the focus of the current manuscript, as we were limited by our smaller sample size for the 16S sequencing.

6. (Line 301 - 303): 'We did observe, however, some evidence to suggest that ferrous bisglycinate significantly increased the relative abundance of Enterobacteriaceae, a finding consistent with other studies in this field'. Are there any references related to ferrous bisglycinate?

After further consideration, we have removed this text “...a finding consistent with other studies in this field”, as the research we were citing was conducted in mice. In their manuscript entitled “Iron supplements modulate colon microbiota composition and potentiate the protective effects of probiotics in dextran sodium sulfate-induced colitis,” Constante et al. (2017) observed that in comparison to ferrous sulfate, mice that received ferrous bisglycinate (50mg/kg) had an increased abundance of members in the Bacteroidales order.

7. The dosage of elemental iron differs greatly between the supplemental forms. How much of iron could end up in the lower gastrointestinal tract?

We updated reference 14 (in L96) to include a new systematic review and meta-analysis of randomized controlled trials with oral ferrous bisglycinate (Fischer et al. 2023). In this review, we cite that it is known that bioavailability is low for iron salts (~5-20%). Based on this data and our dose of ~60 mg elemental iron salts, we can predict that only 3-12 mg iron is absorbed (thus, about 48-57 mg could end up in the GI tract). It is important to note, however, that bioavailability does vary across individuals, so this is a rough estimate.

Specific comments:

8. Figure 3: Please simplify the legend for figure B, merging the representation for week and sample type.

We have simplified the legend as requested.

Reviewer #3

This article is well written and supports the conclusions of the study. I appreciate that the authors published what is essentially negative results and made available all data and code. However, there are several changes that should be made to improve the clarity of the conclusions.

Specifically:

1. Lines 37-42: Remove all instances of "significantly". If something significantly increases, just say increases. If something doesn't significantly increase, then it is OK to say trending. "Significantly" is over-used in scientific literature and unnecessary.

We have removed "significantly" from L37-42.

2. Line 44: Remove "appears to".

We have removed "appears to" from L44.

3. Line 57: Instead of "we hope" use language like This study can facilitate...

L57 now reads "This study can facilitate future research that may inform evidence-based global practice and policy."

4. Line 98-99: Just say, "and the results have been contradictory". No need for "of those that have".

We have removed "of those that have".

5. For Fig. 2: This is a confusing figure. It would make more sense to me if the weeks would be shown alongside each other rather than having "Change in X" on the y-axis (and it should be "Change in X between weeks 0 and 12"). Stars should be explained in the legend. I'm not sure why you compared the time changes between trial arms if you didn't mention it in the text (or the Kruskal-Wallis test for that matter).

We have provided a better figure caption for Figure 2 which accurately describes the analysis.

We have also updated the figure to address the stated areas of confusion.

6. Fig. 3: "Sample Type" in legend should be something like "Trial arm and Week"

We have removed the PCoA plot from Figure 3 and moved it to our supplemental figures. We have changed "Sample Type" to "Trial arm and Week" in the legend as requested.

7. Fig. 4: This is a busy figure and I'm having a hard time understanding why you chose to plot like this. The legend says "Ferrous sulfate increased the relative abundance of Enterococcus and Weissella" but for Weissella it looks like 15 increased while 20 decreased. I'm assuming you mean the average abundance increased? Also, it looks like the colors are switched. The orange lines look like they are decreasing and vice versa for purple. And for Shigella, the green lines definitely have slopes so I don't understand why they would be considered constant. I also don't understand the inclusion of the tables unless you were doing some kind of chi-square or Fisher test.

We have attempted to address the reviewers concerns with Figure 4 by removing the tables and correcting our legend. We have also better explained in the figure caption that the data presented is of the average relative abundance.

8. Fig. 5: Make the color scheme consistent with Fig. 4.

The colour scheme in Figure 4 now matches the colour scheme in Figure 5.

April 28, 2023

Dr. Crystal D Karakochuk
University of British Columbia
Food, Nutrition, and Health
Vancouver
Canada

Re: Spectrum05273-22R1 (The effect of oral iron supplementation on gut microbial composition: A secondary analysis of a double-blind, randomized controlled trial among Cambodian women of reproductive age)

Dear Dr. Crystal D Karakochuk:

Your manuscript has been accepted, and I am forwarding it to the ASM Journals Department for publication. You will be notified when your proofs are ready to be viewed. And please correct the typo as indicated by the last reviewer (on line 629. "trail" -> "trial") in the proofs.

Sincerely,

Zhenjiang Xu
Editor, Microbiology Spectrum

Journals Department
Dear Editor,

The manuscript "**The effect of oral iron supplementation on gut microbial composition: A secondary analysis of a double-blind, randomized controlled trial among Cambodian women of reproductive age**" from Finlayson-Trick and Nearing et al. investigated the impact of oral iron supplementations on gut microbiota. The authors provide valuable insights and have sufficiently addressed comments from the previous reviews.